# Machine Learning for Light Sensor Calibration

**DOI:** 10.3390/s21186259

**Published:** 2021-09-18

**Authors:** Yichao Zhang, Lakitha O. H. Wijeratne, Shawhin Talebi, David J. Lary

**Affiliations:** Hanson Center for Space Sciences, University of Texas at Dallas, Richardson, TX 75080, USA; lhw150030@utdallas.edu (L.O.H.W.); Shawhin.Talebi@utdallas.edu (S.T.)

**Keywords:** spectrophotometer, light sensor, machine learning, neural networks

## Abstract

Sunlight incident on the Earth’s atmosphere is essential for life, and it is the driving force of a host of photo-chemical and environmental processes, such as the radiative heating of the atmosphere. We report the description and application of a physical methodology relative to how an ensemble of very low-cost sensors (with a total cost of <$20, less than 0.5% of the cost of the reference sensor) can be used to provide wavelength resolved irradiance spectra with a resolution of 1 nm between 360–780 nm by calibrating against a reference sensor using machine learning. These low-cost sensor ensembles are calibrated using machine learning and can effectively reproduce the observations made by an NIST calibrated reference instrument (Konica Minolta CL-500A with a cost of around USD 6000). The correlation coefficient between the reference sensor and the calibrated low-cost sensor ensemble has been optimized to have $R^{2}>$ 0.99. Both the circuits used and the code have been made publicly available. By accurately calibrating the low-cost sensors, we are able to distribute a large number of low-cost sensors in a neighborhood scale area. It provides unprecedented spatial and temporal insights into the micro-scale variability of the wavelength resolved irradiance, which is relevant for air quality, environmental and agronomy applications.

## 1. Introduction

Sunlight incident on the Earth’s atmosphere is essential for life, and it is the driving force of a host of photo-chemical and environmental processes (e.g., photosynthesis, photolysis and atmospheric radiative heating). Consequently, models of atmospheric radiative transfer play a key role in modeling atmospheric chemistry and the weather/climate system (e.g., [1,2,3,4,5,6,7,8]). In order to accurately model the surface irradiance, a complete description of both light absorption, light multiple scattering and surface reflection is required in an atmospheric radiative transfer model. For solar zenith angles > 75${}^{\circ }$, this would also need to account for the spherical geometry of the atmosphere [3,4]. The intensity of atmospheric electromagnetic radiation which reaches the Earth’s surface is a strong function of wavelength and the vertical profiles of atmospheric composition and temperature. The vertical profiles of temperature, light scatterers and light absorbers determine the extinction due to absorption and scattering as well as thermal emission.

Atmospheric absorption and multiple-scattering of light both have a significant impact on the surface irradiance (Figure 1). Atmospheric radiative transfer considers the energy transfer of electromagnetic radiation through the atmosphere. The intensity of sunlight, as a function of wavelength, is affected by both the gaseous absorption in the UV and visible portion of the spectrum (including O${}_{3}$, NO${}_{2}$, NO${}_{3}$, HONO and HNO${}_{3}$ [9,10]), by light scattering from air molecules (Rayleigh scattering), from airborne aerosols (Mie scattering) and by thermal emission in the infrared [1,2,11] (Figure 1).

Rayleigh Scattering occurs from gas molecules as the sizes of the molecules are much smaller than the wavelength [12]. The strength of the Rayleigh scattering is proportional to $\lambda^{-4}$, where λ is the wavelength of the radiation. Shorter wavelengths scatter more strongly than longer wavelengths, and this is the reason that the sky is blue. Mie scattering occurs when the size of the scatterers is similar to or greater than the wavelength of the light [13]. In the UV and visible portion of the electromagnetic spectrum that we observed in this study (360–780 nm) the main gaseous absorption of light is due to the ozone in a set of different absorption bands (Hartley 200–300 nm, Huggins 310–360 nm, Chappuis 400–650 nm and Wulf in the near infrared).

### 1.1. Motivation

The goal of this study is to provide an accurately calibrated low-cost wavelength resolved irradiance sensor, which is helpful for biometric pupillometry [14] applications, and is suitable to address the current lack of neighborhood scale real-time solar irradiance data by the provision of very low-cost calibrated measurements. These sensors can be readily deployed at a scale across dense urban environments in order to measure the wavelength resolved irradiance. Sunlight incident on the Earth’s atmosphere is essential for life, and it drives atmospheric photo-chemistry, which is central to understanding urban air quality and the host of associated human health impacts. The World Health Organization (WHO) estimates that, every year, around seven million deaths occur due to exposure to air pollution. Even though the solar irradiance is critical in driving atmospheric photo-chemistry via photolysis, it is marked by a severe paucity of data at the neighborhood scale.

In order to achieve the goal, the first key step is the use of multi-variate non-parametric non-linear machine learning, to accurately calibrate a set of low-cost sensors costing around USD 20 against a NIST calibrated reference instrument. The second step is physically understanding the relative importance of the various factors involved in the calibration. These factors are objectively determined by using explainable machine learning approaches. The plans and circuit diagrams for building these sensors, as well as the calibration code, are publicly available.

### 1.2. Solar Irradiance

The sun is a hot plasma sphere (73% hydrogen, 25% helium and 2% of heavier elements) heated to incandescence by nuclear fusion reactions in the core. The photosphere of the sun has an effective temperature of 5772 K, with an emission spectra close to that of a black body. The electromagnetic energy reaching the top of the Earth’s atmosphere from the sun ranges from 100 nm to 1 mm with a peak at around 500 nm [11].

As the sunlight passes through the Earth’s atmosphere, it is absorbed and scattered by various atmospheric components related to the path length through the atmosphere (Figure 1), which is a function of the solar zenith angle (Figure 2). The solar irradiance incident on the Earth’s surface during the daytime is a function of the Earth’s distance from the sun [15] (varies with season due to the ellipse orbit [16]), the solar zenith angle (can be calculated from latitude, longitude and the local solar time), the vertical profile of atmospheric light scatterers and absorbers and the surface reflectivity.

Typically, a vertical profile through the atmosphere is split up into regions based on the temperature gradient with increasing height, e.g., the troposphere, stratosphere, mesosphere and thermosphere [17]. Significant absorption by ozone of the incoming sunlight occurs in the stratosphere (10–50 km). For typical levels of stratospheric ozone, this light absorption warms the stratosphere and prevents short wave UV at λ< 310 nm (which is harmful to life) from reaching the surface of the Earth.

In a clear sky and without nearby structures such as trees or buildings, solar irradiance is primarily dependent upon some simple factors, such as the temperature of the sun, Earth’s distance and solar zenith angle. However, in a real-world environment, clouds, particulate matter (PM), trees and buildings influence the intensity of solar irradiance on the ground and make it difficult to estimate the irradiance spectrum. Therefore, it is necessary to use high-resolution spectrophotometers to measure the solar spectral irradiance. However, these devices are quite expensive and cannot be widely distributed in large numbers. Thus, we proposed a machine learning method, which works with some low-cost light sensors, in order to achieve competitive performance as well as high-resolution spectrophotometers. The comparisons between the observations of the reference sensor and our machine learning calibrated low-cost sensor ensemble have been shown in Section 4.3 and Section 5. With our machine learning model, we recreated the wavelength-resolved spectrum from 360 nm to 780 nm and obtained an accurate spectrum of atmospheric absorption.

## 2. Measurements and Data Sets

We used two types of light sensors, a NIST calibrated reference sensor (Konica Minolta CL-500A with a cost of ≈USD 6000) and an ensemble of low-cost sensors (Adafruit TSL2591, VEML6075 and AS7262, each costing just a few dollars) to collect the solar irradiance in a same environment. We further calibrated the low-cost sensors against the reference sensor by using machine learning.

The reference Minolta CL-500A provides the irradiance every nm from 360–780 nm, as well as the total illuminance (Figure 3). The codes for collecting the irradiance data from the reference sensor can be found in the following GitHub repository: https://github.com/yichigo/Minolta-Sensor, accessed 18 March 2021.

The various low-cost sensors (Figure 4) sold by open-source hardware company Adafruit Industries are as follows.

AS7262 provides a measurement of the intensity over the broad spectral regions corresponding to “Violet” (450 nm), “Blue” (500 nm), “Green” (550 nm), “Yellow” (570 nm), “Orange” (600 nm) and “Red” (650 nm) light. The circuit design of Adafruit’s AS7262 can be found in https://learn.adafruit.com/adafruit-as7262-6-channel-visible-light-sensor, accessed 22 July 2021.

TSL2591 has a sensitivity to the wavelength range 300–1000 nm, from UV to NIR. It gives the raw counts of “Visible”, “IR (Infrared)” photons and the value of “Lux”. The circuit design of Adafruit’s TSL2591 can be found in https://learn.adafruit.com/adafruit-tsl2591, accessed 22 July 2021.

VEML6075 has a sensitivity relative to the wavelength range of 200–400 nm (UV), where the UVA (UVB) channel has a peak sensitivity at 365 nm (330 nm). It also provides “Visible Compensation” and “IR (Infrared) Compensation” for calculating the UVA (UVB) from the raw counts. In this study, we used the raw counts of UVA, UVB and the compensation values. In addition, VEML6075 provides a calculated “UV index” value, which is negatively correlated with the UV intensity. The circuit design of Adafruit’s VEML6075 can be found in https://learn.adafruit.com/adafruit-veml6075-uva-uvb-uv-index-sensor, accessed 22 July 2021.

The codes for receiving the data from the low-cost sensors are in the following Github repository: https://github.com/mi3nts/UTDNodes/tree/master/firmware/nanoLightUTD, accessed 22 July 2021.

The reference and low-cost sensors were co-located in the same outdoor environment, at the University of Texas at Dallas Waterview Science and Technology Center, 7919 Waterview Parkway, Richardson, TX 75080, from December 2019 to April 2020. They made the observations every 3 s. The data were collected and saved in an NAS hard drive connected in the same network.

## 3. Machine Learning and Workflow

Machine learning was used to calibrate the inputs provided by the suite of low-cost sensors against the reference sensor. Machine learning is a subset of artificial intelligence where we “learn by example”. It optimizes an empirical mathematical model by using examples rather than explicitly programming a deterministic model [18]. This technique is widely used in many areas, such as data mining [19], game strategies [20], healthcare and medical research [21,22,23], computer vision [24,25] and environmental science [26,27]. Machine learning can be divided into three categories: supervised learning, unsupervised learning and reinforcement learning. In this study, we used supervised learning, which trains the model by using a set of examples in a training data set that includes both input features and output targets.

### 3.1. Data Preprocessing

We have collected the data from different sensors. In order to merge these data into a same data set, we resampled the data at every 10 s and merged the different data sources by matching the time. After that, we dropped the NaN values and duplicated data samples. We also calculated the solar zenith angle and solar azimuth angle from the latitude, longitude and UTC time.

The preprocessed data, “MINTS Light Sensor Calibration Dataset”, are publicly available on Zenodo [28].

### 3.2. Input Features and Output Targets

The input features are the preprocessed data given by the low-cost sensors: TSL2591, VEML6075 and AS7262. The output targets are the irradiance bins on 421 wavelengths (360–780 nm) given by the Minolta CL-500A reference sensor. The 14 input features are listed below.


**Features**

**Unit**

**Min**

**Max**

**Mean**

**StDev**
Violet (450 nm)counts02717370.67412.56Blue (500 nm)counts04165528.13611.57Green (550 nm)counts04619546.68664.90Yellow (570 nm)counts04963573.22710.80Orange (600 nm)counts03646411.31519.83Red (650 nm)counts03826425.41545.63IR (Infrared)counts065,53528,500.3527,263.43Visiblecounts051,57312,722.1917,871.05Luxlux−12207.41127.99381.92UVA (Raw)counts040,2962492.983400.20UVB (Raw)counts044,7682682.463686.13Visible Compensationcounts012,045782.751026.57IR (Infrared) Compensationcounts08596456.58699.96UV IndexN/A−1.171.07−0.050.08

The measurement range of the low-cost sensor TSL2591 is not large enough; thus, some of the features may produce unreasonable values. For example, “IR (Infrared)” or “IR (Infrared)” + “Visible” cannot be greater than 65,535 (the maximum of 16-bit integer $2^{16}-1$). Thus, if “IR (Infrared)” is very large, then “Visible” decreases to zero. Another feature, “Lux”, should possess a zero-value minimum; however, in a very bright environment, it produces −1 rather than a large value. Therefore,“Visible” and “Lux” values may be negatively correlated with irradiance, although they should be positively correlated in physics. VEML6075 sensor produces the “UV Index”, which is also negatively correlated with the irradiance, and most of the values are between −0.5 and 0.05.

The output targets given by the Minolta sensor have 421 columns, as listed below.


**Targets**

**Unit**

**Min**

**Max**

**Mean**

**StDev**
Irradiance at 360 nmW/m^2^/nm00.0693380.0104710.011335………………Irradiance at 780 nmW/m^2^/nm00.3376750.0329550.046052

We merged the above features and targets into a single data set. It was randomly shuffled and split up into two portions, 80% of the data were used for training a suite of machine learning algorithms, the remaining 20% was used to independently test the generalization of the machine learning models.

We noticed that there is multi-collinearity between some of the features. For example, the AS7262 sensor’s data “Violet”, “Blue”, “Green” and so on are highly correlated with each other; thus, the machine learning model may focus more on one of them by chance, and their feature importance may also be ranked by chance. Thus, we used principal component analysis (PCA) to remove the multi-collinearity.

### 3.3. Principal Component Analysis (PCA)

The process of PCA can be described as the following: In the N-dimension scaled feature space, we can find a direction that maximizes the variance of the data. We then use that direction as our first principal direction, and we project the data set into an N-1 dimension space by removing the first principal direction. We repeat this process for M times, where M≤N, and obtain a transformed data set in the principal dimensions [29,30].

Typically, people use the PCA technique to reduce the dimensions of the data. However, in this study, we did not reduce the dimension. The input data is still 14-dimension after the PCA process. We only used PCA to remove the multicollinearity between the input features. After training the model, we ranked the feature importances in principle dimensions.

### 3.4. Artificial Neural Network

Artificial Neural Network (ANNs) are one of the many types of machine learning algorithms. The idea central to ANNs is to mimic the neural network found in the human brain in order to solve complex non-linear problems [31].

The left panel of Figure 5 shows an example node (neuron) in a neural network that has four inputs $x_{i}$, four weights $w_{i}$ and one bias. The linear function produces $bias+\sum_{i=1}^{4}x_{i}w_{i}$ and followed by an activation function. A neuron such as this can be used as a linear classifier by optimizing the weights and bias.

Artificial neural networks (ANN) are composed of an input layer, one or more hidden layers and an output layer [32]. As shown in the right panel of Figure 5, each layer has one or more neurons. The input layer receives input features, then feed into the first hidden-layer. The outputs of the first hidden-layer become the inputs to the second hidden-layer and so on. The data passes through all the hidden layers and finally arrives at the output layer. Each node in the hidden layers or output layer has an associated set of weights and bias, and it is followed by an activation function. By using back-propagation [33,34], the gradient of the loss function can be computed with respect to the weights of the network, and the weights can be optimized to fit the model on the training data.

Here, we used a multilayer neuron to calibrate the input features provided by the low-cost light sensors against the data provided by the Minolta reference sensor [35,36,37]. A multilayer neuron (MLP) is a class of feedforward artificial neural network (ANN). We built a three-hidden-layer MLP model, where the sizes of the hidden layers are (64-128-256), and the size of the input (output) layer is 14 (421). We used the ReLu activation function after each hidden layer. The loss function is the mean squared error, and it is optimized by the Adam optimizer. The L2 regularization penalty parameter is $10^{-5}$, and we did not introduce any batch normalization [38] or dropout layer [39,40,41] here.

There are 345,677 date samples. We randomly shuffled the data, then used 80% of the data for training and 20% of the data for testing. In the training data set, the model is actually trained on 90% of the data, and it is valid on the other 10%. We used the standard scaler $x^{\prime }=\left(x-\bar{x}\right)/\sigma$ to scale the input features (before PCA and before ANN model) and output targets based on the training data set, where *x* can be any column of feature or target, $\bar{x}$ is the mean value of *x* and σ is the standard deviation. The size of mini-batches is 200 for stochastic optimizer.

We set the initial learning rate as $10^{-3}$ and trained the model for 40 epochs until the $R^{2}$ validation score was not improved by at least $10^{-4}$ for 10 consecutive epochs. Then, we divided the learning rate by 10 and repeated the process for an additional 12 epochs.

### 3.5. Workflow

In general, the picture of the workflow is shown in Figure 6.

We collected the data form the low-cost sensors and the reference Minolta sensor and performed preprocessing in order to combine and clean the data. Then, we used standard scaler and PCA techniques to generate the scaled PCA input features and scaled output targets for the ANN model. We trained the ANN model on the training data set and explained the model with SHAP values and feature importance. Finally, we tested the performance on the testing data set.

## 4. Machine Learning for Low-Cost Light Sensor Calibration of Wavelength Resolved Irradiance

### 4.1. Whole Spectrum Calibration Model (360–780 nm)

The ANN regression model was trained to calibrate and provide the entire spectrum only from the data provided by the low-cost sensor suite (i.e., to observe if we could reproduce the observations made by a USD 6000 by using sensors costing only a few dollars). For this entire spectrum calibration model, we used 421 neurons as the output layer, and one neuron for each wavelength measured by the Minolta reference sensor.

The upper left panel of Figure 7 shows the scatter diagram showing the performance of this model by comparing the estimated value (*y*-axis) and the actual value (*x*-axis) of the irradiance, from 360 nm to 780 nm, observed by the reference sensor. The coefficient of determination ($R^{2}$) is 0.9987 on the training data and is 0.9983 on the testing data.

The upper right panel shows the quantile–quantile plot, which compares the shape of the probability distribution of our estimates against the shape of the probability distribution of the actual observations. We can observe the distributions of the actual observations, and our estimated values are almost the same above $10^{-3}$ and slightly different below that.

The lower panel shows the $R^{2}$ values for each wavelength between 360 and 780 nm; we observe that that all the coefficient of determination on the testing data are above 0.99. The machine learning model has performed well at all wavelengths.

### 4.2. The Relative Importance of the Machine Learning Inputs

It is very helpful to understand the relative importance of the machine learning inputs. Let us take a look at a couple of approaches that estimate the relative importance of the machine learning inputs in performing the calibration of the low-cost sensors.

#### 4.2.1. Shapley Value: An Explainer of Machine Learning Models

The Shapley (SHAP) value was introduced by Lloyd Shapley in 1951 [42]. It is a game-theoretic approach for calculating the marginal contribution of each player in a cooperative game. In machine learning, we calculated the SHAP value for each data sample on each feature. The SHAP value indicates how much the feature value of a sample raises or decreases the target value.

The processes for calculating the SHAP value are described as follows.

Assume the value function of a teamwork is v(S), where *v* is an arbitrary value function which can be a math function or a machine learning model and *S* is a subset of the players (features) who attended the game, which may contain $x_{0},\;x_{1},\;x_{2},\ldots$. For a given data sample, in order to calculate the value function v(S), only the attended features in *S* use the sample values, while other features use their mean values. The contribution of the player (feature) $x_{i}$ is as follows:(1)$$\phi_{i}=\sum_{S\in \left\{x_{1},x_{2},\ldots ,x_{N}\right\}\backslash \left\{x_{i}\right\}}\frac{|S|!(N-|S|-1)!}{N!}\left(v\left(S\cup \left\{x_{i}\right\}\right)-v\left(S\right)\right)$$
where |S| is the number of the players (features) in *S* and *N* is the total number of the players (features). The term $v\left(S\cup \left\{x_{i}\right\}\right)-v\left(S\right))$ provides the marginal contribution of $x_{i}$ when it joins the game in addition to *S*. The weight $\frac{|S|!(N-|S|-1)!}{N!}$ can be calculated from the permutation of players (features) in which |S|! is the permutation of the players (features) who attended the game before $x_{i}$, (N−|S|−1)! is the permutation of the players who do not attend the game and N! is the permutation of all the players (features). We needed to calculate all the cases of *S* that did not include $x_{i}$ and summed up the weighted marginal contribution of $x_{i}$. Then, we obtained the contribution of $x_{i}$, and that is its SHAP value.

In this study, the inputs and outputs of the ANN machine learning model are scaled by $x^{\prime }=\left(x-\bar{x}\right)/\sigma$, where *x* can be any column in the features or targets, $\bar{x}$ is the mean value of *x* and σ is the corresponding standard deviation. The SHAP values we calculated here are the contributions to the 421 scaled targets (360–780 nm), and we take the averaged SHAP values over these targets.

In order to explain the ANN model, we used the SHAP value to show how each feature contributes to the model’s output, and how each ranks the corresponding feature importance.

We plotted the SHAP values of a random subset of the data points in the following process: normalize the feature values in a color scheme, list different features in the vertical direction and list the SHAP values of each feature in the horizontal direction.

The SHAP values in the left panel of Figure 8 shows how the PCA input features impact the ANN model’s output. For example, the red points on the right side means that if a larger value was the input for this feature, then the output of the model would also increase. We ranked the PCA SHAP values (left panel of Figure 8) by calculating the mean absolute value of the SHAP value. The 0th principle component contributes the most to the models outputs, and other components also contribute a little.

We are able to linearly transform the 0th principle component’s SHAP values to the original features with the first order contribution ratio $\frac{a_{i}\left(x_{i}-\bar{x_{i}}\right)}{\sum_{i}a_{i}\left(x_{i}-\bar{x_{i}}\right)}$, where $x_{i}$ is the *i*-th original feature, $\bar{x_{i}}$ is the mean value of $x_{i}$ and $a_{i}$ is the coefficient of PCA transformation, as shown in the right panel of Figure 8. The original features, namely the red, orange, green, yellow, blue, UVA, UVB and IR (infrared) positively impact the model’s output, while some other features such as the UV index, Visible and Lux features negatively impact the model’s output. Please notice that we only simply use this linear approximation to visualize the main part (from 0th principle component) of the contributions of the original features. However, we should not sum up these original features’ SHAP values by combining all the principle components because the original features have strong multi-collinearity.

#### 4.2.2. Feature Importance

For machine learning models such as ANN, we can calculate feature importance from the mean absolute value of the Shapley (SHAP) values for each feature.

In our MLP model for calibrating the whole spectrum, we calculate the feature importance from the SHAP values, as shown in Figure 9. We used the red (blue) color to indicate a feature that positively (negatively) impacts the model. The positive/negative impact means that if a feature’s value increases with the other features remaining unchanged, then the output value of the model is more likely to increase/decrease. For example, in the Figure 8 of SHAP values, the 0th principle component has a positive impact on the model’s output since the red (blue) points, with high (low) feature values, are on the right (left) side, which raises (decreases) the model’s output. In most of the models, the sign of positive and negative can also be easily calculated from the sign of the correlation coefficient between the feature and the target. The feature importances are ranked in descending order, and we are able to figure out which features are important for predicting our output targets. We note that most of the variables provided by the low-cost light sensor suite provide useful information.

Now, we have observed that machine learning can provide an effective calibration of the low-cost sensors, and we are familiar with the relative importance of the input parameters; let us use this calibration to examine the temporal variability that was measured.

### 4.3. Applying the Calibration to Provide an Irradiance Spectrum

We picked at random some time periods from the testing data set and used our neural network model to provide the full spectral irradiance from 360 nm to 780 nm by only using the data provided by the low-cost USD 20 sensor ensemble (Figure 10). We estimated the full spectral irradiance (middle panel) from the low-cost sensor ensemble and compared this with the observed value from the reference NIST calibrated sensor (upper panel). Our ANN model successfully reproduced the high resolution spectrum by only using the data from the low-cost sensors.

We can clearly observe the role of the variable weather conditions, such as changing cloudiness. When the direct normal irradiation (DNI) is stopped by the surrounding buildings, the photons reflected by the clouds increase the total intensity of irradiance. Furthermore, as the light scattering is wavelength dependent, the shape of the spectrum changes on a cloudy day. It was sunny on 5 March 2020. Note the higher spectral irradiance at the shorter wavelengths, indicating a blue sky day. By comparison, it was cloudy on 29 March 2020. Note how the spectral irradiances from the blue to red portion of the spectrum are much more similar than on March 5; the sky was closer to white than blue. Our ANN performed well for both clear sky and cloudy conditions, and the data from the low-cost sensors could be used to effectively reproduce the full spectra.

In the bottom panel of Figure 10, we overlay the actual solar spectrum and estimated solar spectrum for both sunny (5 March 2020) and cloudy (29 March 2020) cases. Furthermore, the figure displays the atmospheric absorption spectrum, including nitric oxide (NO) at 429 nm, oxygen ($O_{2}$) at 688 nm and 762 nm and water vapor ($H_{2}O$) from 720 nm to 730 nm [43]. Using data from the low-cost sensors, our machine learning model correctly obtained all of these absorption bands and showed the potential for detecting changes of atmospheric components.

Our ANN model was trained on the MINTS light sensor calibration data set [28], the codes for training the model and generating all the figures can be found in the following Github repository [44]: https://github.com/yichigo/Light-Sensors-Calibration, accessed 25 August 2021.

## 5. The Observed Diurnal Variation in Wavelength Resolved Irradiance

Figure 11 shows the observations used in the testing data set for the entire three month period. As expected, the solar zenith is a key factor in determining solar irradiance. Secondly, we note that, on cloudy days, the multiple scattering of light from airborne aerosols and clouds coupled with the surface reflection of the scattered light increases the surface irradiance substantially (Figure 11). In Figure 11, we observe that the surface irradiance observed by the reference sensor and that is estimated by the low-cost ensemble of sensors calibrated using machine learning agree very well; the blue and red points overlay one another so precisely that they produce the appearance of magenta points. In each panel, the solid curves close to the bottom shows the averaged irradiance for the sunny days, and the cloudy days produce a higher value of irradiance due to the trapping of photons by multiple scattering from the clouds and surface reflection of the scattered light.

Figure 12 compares the wavelength (*y*-axis) and UTC time (*x*-axis) resolved daily spectra for a sunny day on the left, and for a cloudy day on the right. In both cases, we can observe the key role that both the solar zenith angle and the cloudiness plays in determining the intensity of sunlight at the surface of the Earth.

We can observe that the surface irradiance wavelength distribution is a strong function of the conditions and that atmospheric multiple scattering of sun light plays a substantial role. For a sunny day, the solar irradiance spectrum peaks in the violet and blue part of the spectrum and is close to that of a black body at around 5772 K. On a cloudy day the multiple light scattering and surface reflection traps photons, thereby enhancing the intensity at the longer wavelengths. The cloud water drops result in Mie scattering of the sunlight. Thus, the diffuse horizontal irradiance (DHI) is actually greater on a cloudy day than it is on a sunny day. This is very evident when we take a snapshot at an instant in time and examine the shape of the spectrum as a function of wavelength and compare a sunny and cloudy day (Figure 13).

The sunny day irradiance spectrum is close to that of a black body at around 5772 K. The partly cloudy spectrum has enhanced intensity, particularly at longer wavelengths due to Mie scattering in the clouds. The Mie scattering on the clouds generates greater diffuse horizontal irradiance (DHI) and makes the sky brighter.

## 6. Conclusions

A neural network algorithm was able to effectively calibrate an ensemble of low-cost light sensors and generate a high resolution wavelength resolved solar irradiance spectrum. The roles of the solar illumination geometry (such as the solar zenith angle) and the weather conditions (such as the cloudiness) were clearly evident. These low-cost light sensor packages are currently being deployed across cities in the Dallas Fort Worth (DFW) area for characterizing both the temporal and spatial scales of solar irradiance. All the sensor circuit designs and calibration codes have been made open source.

## Figures and Tables

**Figure 1 sensors-21-06259-f001:**
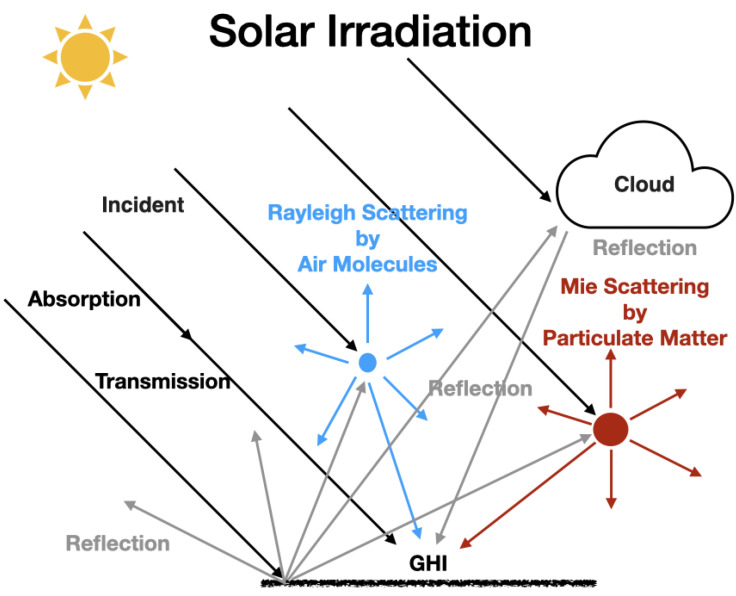
Solar irradiance on the Earth’s Surface.

**Figure 2 sensors-21-06259-f002:**
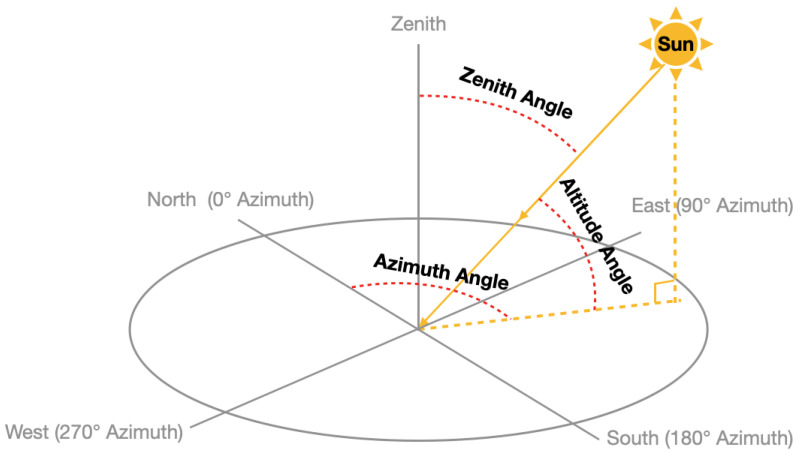
Schematic depicting the solar zenith angle, solar altitude angle and solar azimuth angle.

**Figure 3 sensors-21-06259-f003:**
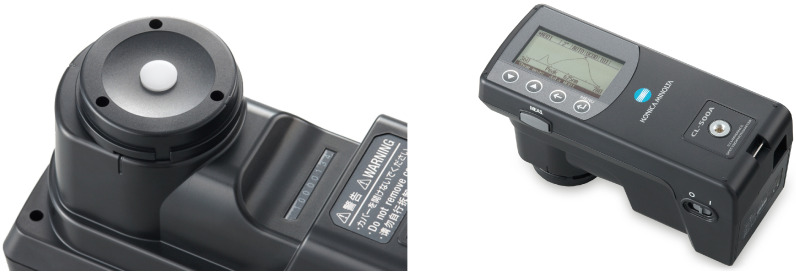
Konica Minolta CL-500A Illuminance Spectrophotometer measures the wavelength range of 360–780 nm.

**Figure 4 sensors-21-06259-f004:**
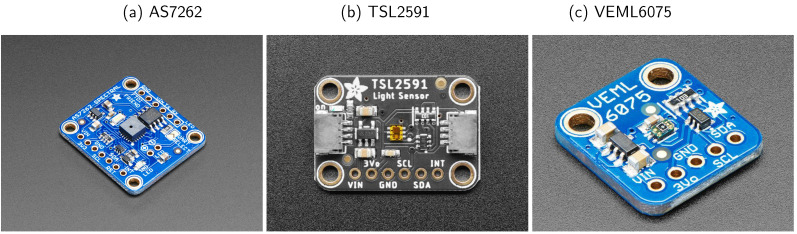
Low-cost light sensors: TSL2591, VEML6075 and AS7262 sold by the open-source hardware company, Adafruit Industries.

**Figure 5 sensors-21-06259-f005:**
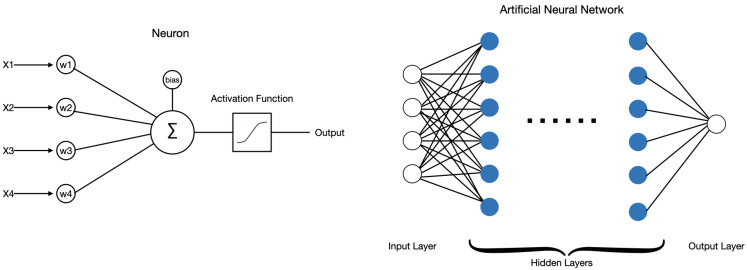
The left panel shows an example node in a neural network, where $w_{i}$ is the weights for each input $x_{i}$, and the linear function produces $bias+\sum_{i}x_{i}w_{i}$ that is passed to an activation function. The right panel shows an example of artificial neural network (ANN) with single output, where the blue nodes in hidden layers and the white node in the output layer are neurons.

**Figure 6 sensors-21-06259-f006:**
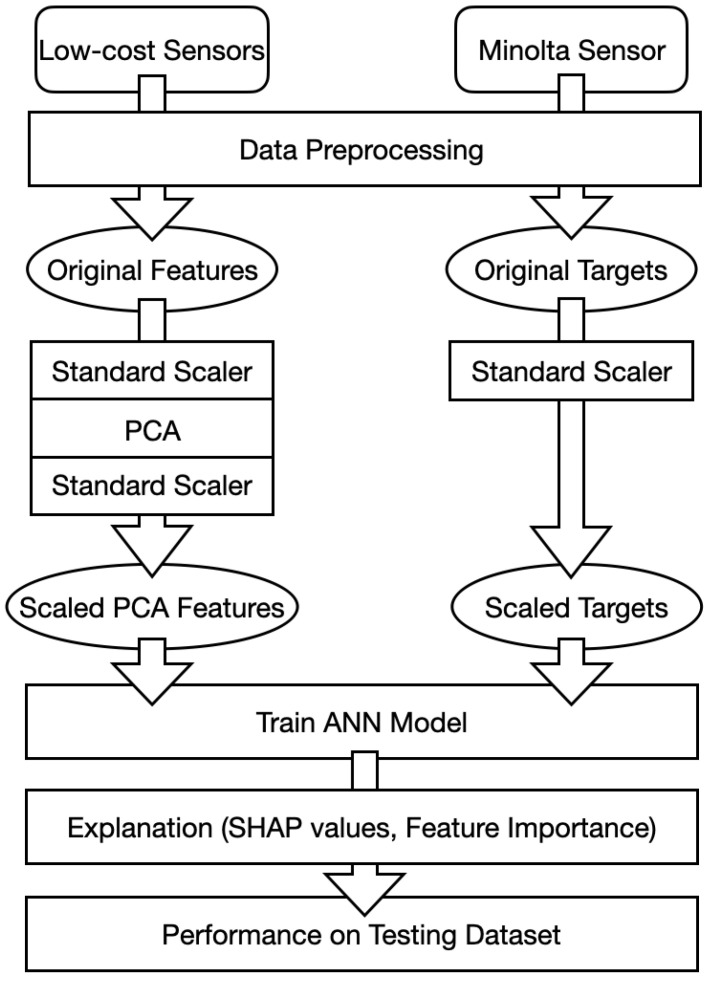
Workflow of the light sensor calibration.

**Figure 7 sensors-21-06259-f007:**
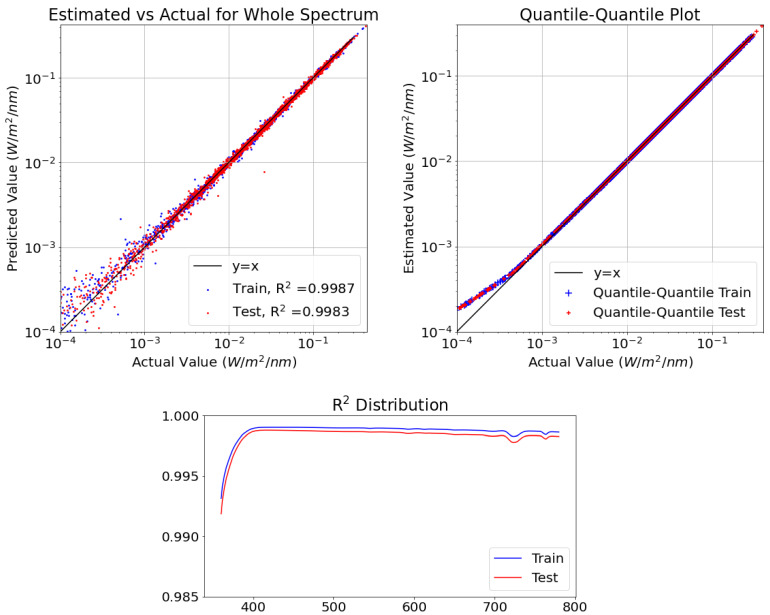
Whole spectrum model performance and the R2 scores on different wavelengths.

**Figure 8 sensors-21-06259-f008:**
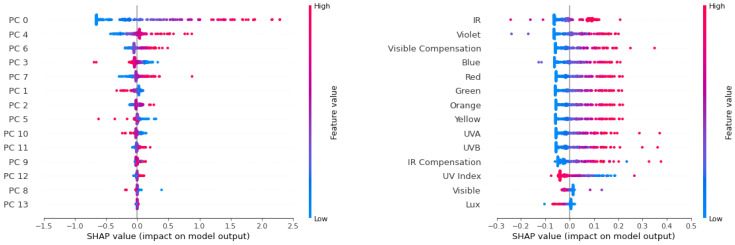
SHAP value of our MLP model, which calibrates the whole spectrum, shows how the PCA features impact the models’ output. The red (blue) color denotes the high(low) value of each feature and the position of point on the *x* direction shows the impact on the target. The left panel shows the principle components’ SHAP values, and the right panel shows the original features’ SHAP values from the 0th principle component by using linear approximation.

**Figure 9 sensors-21-06259-f009:**
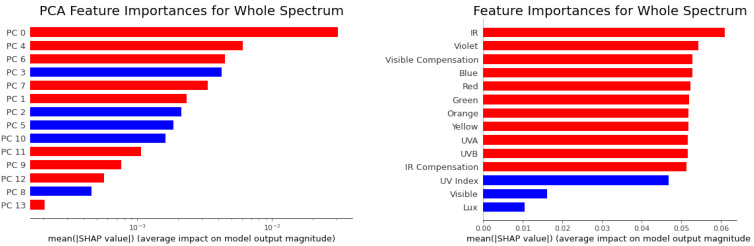
(**Left**) PCA feature importance of our MLP model for whole spectrum calibration; we used a $log_{10}$ scale in the x direction. (**Right**) The original feature importances from the linearly splitting of the 0th principle component’s SHAP values. A feature with red (blue) color has positive (negative) impact on the model’s output.

**Figure 10 sensors-21-06259-f010:**
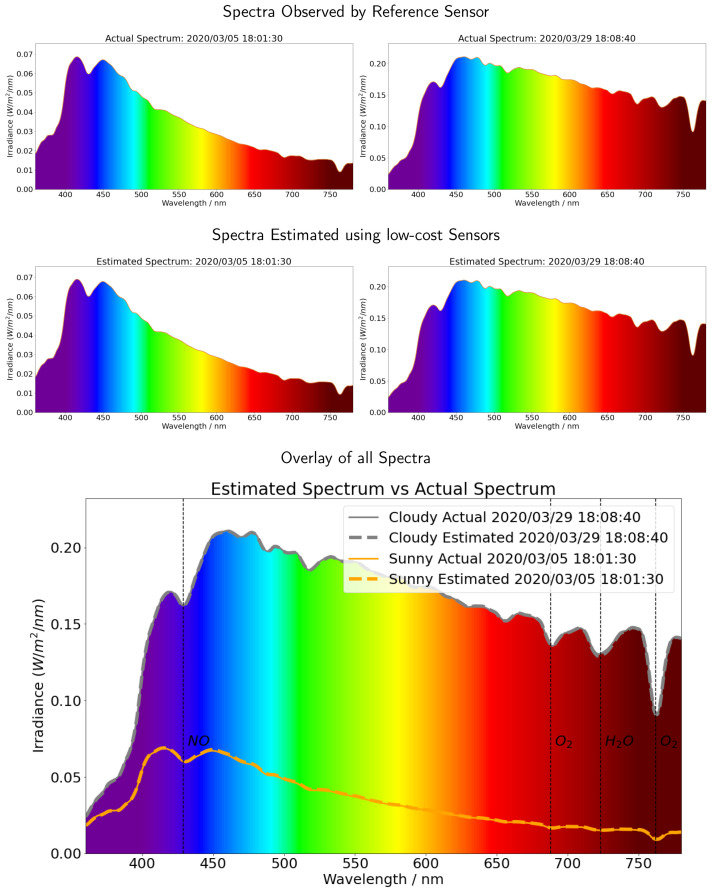
The full spectrum intensity (middle panel) predicted using the data from the low-cost sensors, and the full spectrum ANN compared with the spectra observed by the reference sensor (upper panel). The daily spectra over the entire day have been shown in Section 5. We can clearly observe the role of the variable weather conditions, such as changing cloudiness. It was sunny on 5 March 2020, and this can be observed by noting the higher spectral irradiance at the shorter wavelengths, indicating a blue sky day. In comparison, it was cloudy on 29 March 2020. Note how the spectral irradiances from the blue to red portion of the spectrum are much more similar than on March 5; the sky was closer to white than blue. Our ANN performed well for both clear sky and cloudy conditions, and the data from the low-cost sensors could be used to effectively reproduce the full spectra, including the atmospheric absorption bands.

**Figure 11 sensors-21-06259-f011:**
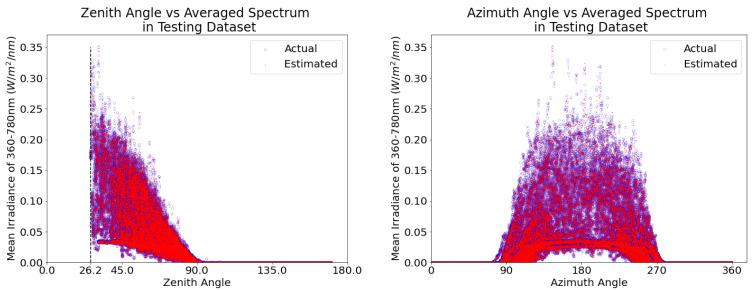
The mean irradiance from 360–780 nm as a function of solar zenith angle (**left panel**) and solar azimuth angle (**right panel**) from the testing data set for the entire three month period. The blue circles show the actual irradiance; the overlaid red points show the machine learning estimates. We observe that the surface irradiance observed by the reference sensor and that is estimated by the low-cost ensemble of sensors calibrated using machine learning agree very well; the blue and red points overlay one another so precisely that they produce the appearance of magenta points. In each panel, the solid curves close to the bottom shows the averaged irradiance for the sunny days, and the cloudy days produce higher values of irradiance due to the trapping of photons by multiple scattering from the clouds and surface reflection of the scattered light.

**Figure 12 sensors-21-06259-f012:**
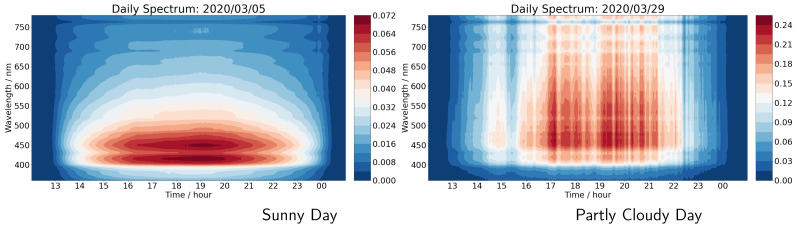
A comparison of the wavelength (*y*-axis) and UTC time (*x*-axis) resolved daily spectra for a sunny day on the left and for a cloudy day on the right. The color denotes the solar irradiance in unit W/m${}^{2}$/nm. Both spectra were collected using the Minolta CL-500A Illuminance Spectrophotometer.

**Figure 13 sensors-21-06259-f013:**
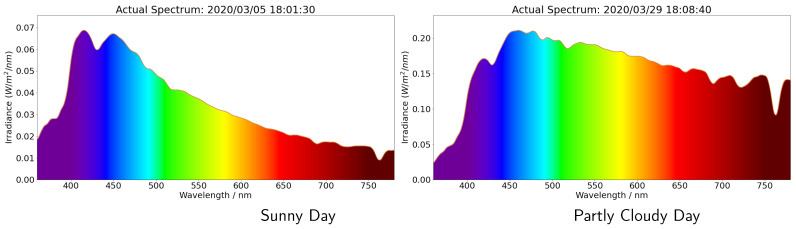
Spectra of solar irradiance measured by Minolta CL-500A Illuminance Spectrophotometer on a certain time, where x value is the wavelength, y value is the intensity of irradiance in unit W/m${}^{2}$/nm and the color denotes the visible color of the corresponding wavelength.

## Data Availability

Not applicable.

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
