# Peer review of "Machine Learning for Light Sensor Calibration"

_sensors, 2021, doi:10.3390/s21186259_

Round 1

Reviewer 1 Report

The comments are provided in the attached document

Author Response

Thank you for inviting us to revise our manuscript “Machine Learning for Light Sensor Calibration”. As well as appreciating your time and efforts, both of you have provided thoughtful suggestions on how we can improve our paper. Therefore, we re- submit our work with great pleasure for further consideration. We have incorporated changes based on your detailed suggestions. You can also open the overleaf link of the revision through https://www.overleaf.com/read/wrcpmsxvrfmw. For reading convenience, the big changes in the manuscript have been colored in red. We hope that all the issues and you concerns have been adequately addressed by our edits and answers below.

Author Response

(The authors gave the same response as above.)

Round 2

Reviewer 2 Report

The manuscript got much improved after revision. Two minus suggestions below:

1.  Line196, "...trained the model for 40 iterations until the R2...", I quite doubt that only 40 iterations can make model cost converge and reach global minimum for 345,677 data sample. Also, the batch size is 200, for 40 iterations, only 8000 data samples were actually used in your training. That is impossible to make your model converge and get optimum.  Did you mean 40 epochs ?

2. The ANN method used in the following paper is quite similar with your research. The paper also included L2 regularization and batch normalization. It is good idea to add it as a reference to assist your ANN description in section 3.4.    

Liang, X.; Liu, Q. Applying Deep Learning to Clear-Sky Radiance Simulation for VIIRS with Community Radiative Transfer Model—Part 2: Model Training, Test and validation. Remote Sens. 2020, 12, 3825.

Author Response

Dear Managing Editor,

We sincerely appreciate your follow-up review of the manuscript “Machine Learning for Light Sensor Calibration”. Thank you very much for your time and efforts. Our work is submitted again with great pleasure for your consideration. We have implemented your suggestions as shown in the attached file.

Best,

Yichao Zhang
